# Enhancing the Solubility and Dissolution of Apigenin: Solid Dispersions Approach

**DOI:** 10.3390/ijms26020566

**Published:** 2025-01-10

**Authors:** Natalia Rosiak, Ewa Tykarska, Andrzej Miklaszewski, Robert Pietrzak, Judyta Cielecka-Piontek

**Affiliations:** 1Department of Pharmacognosy and Biomaterials, Faculty of Pharmacy, Poznan University of Medical Sciences, 3 Rokietnicka St., 60-806 Poznan, Poland; nrosiak@ump.edu.pl; 2Department of Chemical Technology of Drugs, Poznan University of Medical Sciences, 3 Rokietnicka St., 60-806 Poznan, Poland; etykarsk@ump.edu.pl; 3Faculty of Materials Engineering and Technical Physics, Institute of Materials Science and Engineering, Poznan University of Technology, 60-965 Poznan, Poland; andrzej.miklaszewski@put.poznan.pl; 4Faculty of Chemistry, Adam Mickiewicz University, 8 Uniwersytetu Poznańskiego St., 61-614 Poznan, Poland; pietrob@amu.edu.pl

**Keywords:** apigenin, solid dispersion, solubility, FT-IR

## Abstract

Apigenin (APG), a bioactive flavonoid with promising therapeutic potential, suffers from poor water solubility, which limits its bioavailability. To address this, solid dispersions of APG were prepared using ball milling with sodium alginate (SA), Pluronic^®^ F-68 (PLU68), Pluronic^®^ F-127 (PLU127), PVP K30, and PVP VA64 as polymeric excipients. These dispersions were screened for apparent solubility in water and buffers with pH 1.2, 5.5, and 6.8. Based on improved solubility after 60 min, APG–PLU68 and APG–PLU127 dispersions were selected for further study. DSC and FT-IR analysis confirmed molecular interactions between APG and the polymer matrices, contributing to enhanced solubility and dissolution rates. Dissolution rate studies showed that APG–PLU127 achieved 100% solubility at pH 6.8, suggesting its potential use in environments such as the small intestine. Additionally, APG–PLU127 exhibited 84.3% solubility at pH 1.2, indicating potential for solid oral dosage forms, where APG could be absorbed in the acidic conditions of the stomach. The stability study confirmed that storage for one year under ambient conditions does not cause chemical degradation but affects the physical state and solubility of the dispersion. Antioxidant activity was assessed using the ABTS assay. Freshly obtained APG–PLU127 showed 68.1% ± 1.94% activity, whereas APG–PLU127 stored for one year under ambient conditions exhibited 66.2% ± 1.62% (significant difference, *p* < 0.05). The difference was related to a slight decrease in the solubility of APG in the solid dispersion (T0 = 252 ± 1 μg∙mL^−1^, T1 = 246 ± 1 μg∙mL^−1^). The findings demonstrate the superior performance of PLU127 as a carrier for enhancing the solubility, release, and antioxidant activity of APG.

## 1. Introduction

Modern pharmacology and medicine face numerous challenges, among which one of the most critical is improving the bioavailability of drugs. A key factor influencing bioavailability is the solubility of a drug in water, as only the dissolved fraction of the active substance can be absorbed by the body and exert the desired therapeutic effects [1,2]. The issue of poor solubility affects nearly 90% of newly developed drugs, posing a significant challenge for the pharmaceutical industry [3]. Promising results can be achieved through chemical and physical modifications of the medicinal compound, as well as by incorporating appropriate excipients into the final drug formulation. Chemical methods include salt formation, complexation, and prodrug design, while physical methods focus on drug delivery systems—modifications that alter the physical form of the substance without changing its molecular structure [4,5,6]. These methods primarily include particle size reduction, amorphization, and the creation of solid dispersions or cocrystals [7].

Apigenin (APG, Figure 1), a flavone, demonstrates significant lipophilicity, which allows it to penetrate biological membranes; however, its low solubility in water (1.35 µg/mL) [8] limits its application despite many biological properties, including anti-inflammatory, antioxidant, and antibacterial effects [9,10].

Despite the potential beneficial therapeutic properties of APG, it is not widely used in medicine due to limitations resulting from low bioavailability. APG belongs to Class II of the BCS system, which means that despite good permeability through membranes, its activity is limited by water solubility [11]. For this reason, attempts are being made to increase bioavailability by creating APG delivery systems. Table 1 summarizes the APG delivery systems described so far in the scientific literature.

Solid dispersions in polymer matrices offer a promising approach to enhancing the solubility and bioavailability of medicinal compounds [12,13,14,15,16]. Therefore, in this study, we aim to create a solid dispersion of APG with sodium alginate, Pluronic^®^ F-68, Pluronic F-127^®^, PVP K30, and PVP VA64. The ball milling method (BM) was used to obtain them. Our previous study confirmed that BM enables effective particle size reduction, resulting in very fine drug particles with increased specific surface area [28]. Additionally, the mechanical action of the ball mill ensures high dispersion uniformity, yielding a product with more controlled physicochemical properties. Achieving similar uniformity can be challenging with other techniques, such as spray drying. Another significant advantage of the ball mill is that it operates without the need for high temperatures. This eliminates the risk of thermal degradation of the drug, which is sensitive to heat, unlike methods such as spray drying that may lead to the decomposition of bioactive compounds. Furthermore, the ball mill is relatively simple to operate and has low operating costs. The milling process also enhances the stability of the dispersion. Milling can be conducted in the presence of stabilizers or carriers, which help prevent recrystallization or aggregation of active pharmaceutical ingredients (APIs), making the ball mill particularly effective for preparing stable dispersions. Moreover, the process is highly flexible, allowing for adjustments to parameters such as milling speed, duration, and ball size to optimize the final product properties. In comparison to other techniques, such as precipitation or ultrasonic methods, the ball mill often provides superior control over particle size and distribution, translating to better functional properties of API dispersions. Moreover, dry milling is considered an environmentally friendly method for preparing dispersions. One of the primary environmental benefits of this technique is that it does not require the use of solvents or other chemical agents, reducing the environmental impact associated with solvent disposal and contamination. This solvent-free process eliminates the need for hazardous or toxic chemicals, making it a safer and more sustainable alternative compared to liquid-based techniques, such as solvent evaporation or precipitation, which typically involve the use of large amounts of solvents [29,30,31].

These advantages highlight the ball mill as a versatile and efficient method for preparing APG dispersions for various applications. The results of this research may contribute to the development of new pharmaceutical formulations containing APG, which could be effectively used in the treatment of multiple diseases. Additionally, this work represents a contribution to the advancement and application of mechanochemical pharmaceutical technologies in line with sustainable development principles, which is of critical importance in the context of global environmental challenges.

## 2. Results and Discussion

Solid dispersions of apigenin (APG) were prepared by the dry milling method using sodium alginate (SA), Pluronic^®^ F-68 (PLU-68), Pluronic^®^ F-127 (PLU-127), PVP K30 (PVP30), and PVP VA64 (PVPVA64) as carrier matrices. The structures of the carriers are presented in Figure 2.

SA is the sodium salt of alginic acid, which is a polymer composed of D-mannuronic and L-guluronic acid units linked in a chain by a β-1,4-glycosidic bond. This substance is widely used in the food, pharmaceutical, and cosmetic industries due to its stabilizing, thickening, gelling, and emulsifying properties [32]. SA has been recognized by the U.S. Food and Drug Administration (FDA) and the World Health Organization (WHO) as a safe food additive. Due to the lack of absorption, SA can also be used by children and breastfeeding women [33]. PLU is the commonly used trade name for poloxamer—a copolymer made of poly(ethylene oxide, PEO) and poly(propylene oxide, PPO) mer units. The PEO group exhibits hydrophilic properties, while the PPO group is lipophilic. Individual types of PLU differ in their state of aggregation, the ratio of the mer units present, and their molecular weight, and all this information is included in the name of a specific type of poloxamer. The state of aggregation is marked with the letters L, P, and F, which mean a liquid, paste, and solid, respectively. Meanwhile, the numbers attached to this letter carry information about the molecular weight and the ratio of PEO and PPO groups. The last digit multiplied by 10 defines the percentage of hydrophilic PEO groups, and the number preceding it, multiplied by 300, represents the molecular weight of the hydrophobic groups. For example, the name Pluronic^®^ F-68 denotes a solid substance with a hydrophilic group content of 80% and a hydrophobic part mass of 1800 Da. Pluronic F-68 (PLU68) and Pluronic^®^ F-127 (PLU127) are the only poloxamers to have received FDA approval for safety and biocompatibility [33]. PVP30 is a polymer—polyvinylpyrrolidone, and PVP VA64 is a copolymer of vinylpyrrolidone and vinyl acetate. The hydrocarbon chain gives the molecule a lipophilic character, and the presence of pyrrolidone and an acetate group gives a hydrophilic character; hence, these compounds are classified as amphiphilic. Animal studies confirm that PVP does not cause acute or chronic poisoning in the body. Moreover, these are substances that are compatible with tissues and are non-irritating, both when administered orally and locally. The safety profile has enabled the wide use of PVP in pharmacy. As an excipient, PVP can prevent recrystallization and act as a binding, coating, lubricant, carrier, stabilizer, and solubilizer agent. It can be found in tablets, capsules, granules, pellets, solutions, suspensions, ointments, and parenteral, ophthalmic, and nasal preparations [34].

In our study, dispersions obtained were subsequently analyzed using X-ray powder diffraction (XRPD) to assess their solid-state structure. The formulations showing the most significant improvement in apparent solubility in water and buffers of pH 1.2, 5.5, and 6.8 were selected for further investigation. For these chosen samples, identity confirmation was conducted using differential scanning calorimetry (DSC) and Fourier-transform infrared spectroscopy (FT-IR) to evaluate potential APG–polymer interactions, which were responsible for the enhanced solubility and dissolution rate of APG.

Analysis using XRPD was conducted on pure APG, APG after a 60-min milling process (APG_60’), APG–polymer physical mixtures (PM), and APG–polymer solid dispersions (BM). This investigation aims to evaluate the solid-state characteristics of APG before and after the milling process, as well as in its various formulations. By comparing the XRPD patterns of these samples, we can gain insights into the effects of mechanical treatment and the incorporation of excipients on the structure and overall properties of APG. Understanding these changes is crucial for assessing how they might influence the solubility and bioavailability of APG in pharmaceutical applications.

Figure 3 shows the diffraction patterns of pure APG and APG_60’.

The diffraction pattern of pure APG exhibits several sharp peaks at 2θ angles: 7.0°, 9.9°, 11.1°, 14.1°, 14.9°, 15.8°, 18.1°, 23.7°, 25.1°, and 26.1°, confirming the crystalline structure of the compound. Additionally, peaks of lower intensity are observed at 2θ angles: 20.0°, 20.7°, 26.9°, 28.5°, 31.8°, 33.9°, and 36.0°. This diffraction pattern aligns with previously published results [13,14,15,35]. After the milling process, a decrease in peak intensity is observed. Nonetheless, the presence of Bragg peaks in the APG_60’ diffraction pattern confirms that the compound retains its crystalline form.

Figure 4, Figure 5 and Figure 6 show the diffractograms of APG, polymers (SA, PLU68, PLU127, PVP30, PVPVA64), APG–polymer physical mixtures, and APG–polymer solid dispersions. APG peaks are found in the diffractograms of all physical mixtures, which proves the presence of crystalline APG.

The diffractogram of pure SA (Figure 4, red line) shows a broad “halo” effect with local maxima at angles of 13.5° 2θ and 21.7° 2θ, which indicates the amorphous form of the polymer. Similar diffractograms of this substance have already been described in the literature [36,37]. In the diffractogram of APG–SA BM, the characteristic peaks of APG diminish. Moreover, the local maxima of SA are flattened. These observations suggest the transition of APG into the amorphous form in the APG–SA two-component system after the ball milling process.

The diffraction pattern of pure PLU68 and PLU127 shows two intense Bragg peaks at 2θ angles of 19.2° and 23.3°, which indicates the semi-crystalline form of the polymer (Figure 5, red line). Their position is consistent with data published in the scientific literature [38,39,40,41].

According to Shatalova et al. [42], the layered structure of Pluronic^®^, similar to that of many semicrystalline polymers, features alternating crystalline and amorphous layers. The crystalline regions are composed of PEO layers, while the amorphous regions include both PPO and PEO. The results obtained for APG–PLU68 BM and APG–PLU127 BM indicate that APG is fully dispersed in the Pluronic^®^ matrix.

The diffraction patterns of pure PVP30 and PVPVA64 are consistent with the literature and demonstrate the amorphous nature of these polymers (Figure 6) [43,44]. After milling, the Bragg peaks from APG present in diffractograms of physical mixtures of APG–PVP30 and APG–PVPVA64 diminished, indicating the dispersion of APG in a polymer matrix.

The study of apparent solubility was conducted to assess the impact of various carriers on enhancing the solubility of APG in water as well as in buffers with pH levels of 6.8, 5.5, and 1.2. Table 2 presents the results of the apparent solubility study of APG and APG solid dispersions.

This investigation aimed to explore how different excipients could overcome the poor aqueous solubility of APG, which is a major limitation in its therapeutic application. By examining solubility in a range of pH conditions, we sought to simulate different physiological environments, such as the gastrointestinal tract, where drug absorption occurs. The findings from this solubility study will serve as a foundation for developing optimized drug formulations that can improve the delivery and therapeutic effectiveness of APG in the future.

The detection and quantification limits were also determined to be 0.1 μg∙mL^−1^ and 0.04 μg∙mL^−1^, respectively. Pure APG gave results below the detection limit at both time points and in all media. The APG–SA and APG–PVPVA64 dispersions also gave results below the detection limit in all media. The APG–PVP30 dispersion was below the detection limit in water and pH 6.8, while at pH 1.2 and pH 5.5, a slight increase in solubility was observed after 15 min and a decrease after 60 min. The best improvement in APG solubility was achieved by solid dispersions of PLU68 and PLU127, with PLU127 showing significantly stronger solubilizing properties. A comparison of the apparent solubility results obtained at two time points indicated that APG–PLU127 showed an increase in solubility in all media after 60 min, while APG–PLU68 showed an increase in all media except in water. The superior performance of APG–PLU127 in enhancing solubility can be attributed to the differences in the structural properties of PLU68 and PLU127, specifically the hydrophilic–lipophilic balance (HLB) and the ratio of polyoxyethylene (PEO) to polyoxypropylene (PPO) units. PLU127, characterized by a lower PEO/PPO ratio, possesses greater hydrophobicity compared to PLU68. This structural feature is crucial for interactions with lipophilic compounds such as APG. The stronger hydrophobic interactions between the PPO core of PLU127 and the APG molecules result in more effective solubilization. A study [45] confirmed that PLU127 forms micelles with a well-defined hydrophobic PPO core and a hydrated PEO shell. This configuration allows for efficient encapsulation of lipophilic molecules while maintaining solubility in aqueous environments. The PEO shell stabilizes the micelles by providing steric hindrance, preventing premature aggregation or dissociation. In contrast, the higher PEO content in PLU68 increases its hydrophilicity, leading to weaker interactions with APG and reduced solubilization efficiency. An optimal balance between hydrophobicity and hydrophilicity is critical for achieving both effective solubilization and micellar stability. Dispersions of hydrophobic API with polymers of very high hydrophobicity, such as PLU68, are prone to instability in aqueous media because the polymer molecules aggregate to form large lamellar structures. However, PLU127 strikes a balance, exhibiting sufficient hydrophobicity to interact strongly with APG while maintaining micellar stability in water. This is reflected in its HLB value, which is better suited for solubilizing lipophilic substances compared to the more hydrophilic PLU68. Additionally, the compatibility of PLU127’s PPO core with APG molecules likely enhances its performance. The stronger hydrophobic interactions facilitate tighter packing of APG within the micelle core, minimizing the risk of APG precipitation or crystallization in the aqueous phase.

Based on the apparent solubility studies, APG–PLU68 and APG–PLU127 were selected for release profile studies in water and buffers at pH 1.2, 5.5, and 6.8. The study aimed to characterize the effect of amorphization on APG release in specific media. The release profiles of APG from APG–PLU68 and APG–PLU127 solid dispersions within 6 h in the tested media are presented in Appendix A. The APG–PLU68 achieved the best result after 6 h at pH 6.8—it was 21.0%. Next were pH 5.5 (13.5%), pH 1.2 (7.4%), and pure water (1.2%). Much higher results were achieved in the system with PLU127, where the highest substance released was 100% at pH 6.8. The remaining ones were 84.3% at pH 1.2, 47.0% in pure water, and 40.9% at pH 5.5. For each medium, it can be observed that a larger percentage of the substance was released from the APG–PLU127.

The release profiles of APG from PLU127 and PLU68 dispersions in water (Figure 7) highlight the superior performance of PLU127.

A rapid release phase was observed for PLU127, with approximately 35% of APG dissolved within the first 120 min. This was followed by a slower phase, reaching a plateau at about 45% after 240 min. In contrast, the PLU68 dispersion showed negligible dissolution throughout the experimental period, with the percentage of released APG remaining close to zero. These differences are attributed to the distinct physicochemical properties of the polymers. The amphiphilic nature of PLU127 allows for micelle formation, enhancing the solubilization and dispersibility of APG, whereas the weaker micellization of PLU68 results in poor solubilization.

Under acidic conditions—pH 1.2 (Figure 8), PLU127 achieved rapid APG release, with over 40% dissolved within 30 min, reaching a plateau of approximately 84% by 300 min.

The biphasic release profile underscores its capacity to enhance solubilization through micelle formation and improved wettability. In contrast, PLU68 exhibited limited dissolution, with less than 10% of APG dissolved after 360 min, confirming its inefficacy as a solubilizing agent under gastric conditions. At pH 5.5 (Figure 9), PLU127 displayed a similar biphasic release, dissolving approximately 25% of APG within 30 min and reaching a maximum of around 40% by 360 min.

The observed dynamics suggest effective solubilization through micellar interactions. Meanwhile, PLU68 again exhibited minimal release, with only about 10% APG dissolved, progressing linearly throughout the experiment.

At pH 6.8 (Figure 10), the solubility of APG was significantly improved with PLU127. The dispersion dissolved approximately 40% of APG within 60 min, reaching nearly 100% dissolution in 360 min.

The enhanced performance is likely due to more stable micelle formation and stronger interactions between APG and the polymer. In contrast, PLU68 showed only a slight improvement, with 15% of APG dissolved after 360 min, maintaining a slow and linear release profile.

Domínguez-Delgado et al. [46] indicate that PLU introduced into the body via routes other than dermal exposure is rapidly cleared from the body, suggesting that there would be no risk of reproductive and/or developmental toxicity. For this reason, APG–PLU127 solid dispersion, characterized by 100% release of APG in pH 6.8, suggests that the dispersion demonstrates excellent solubility under conditions representative of the intestinal environment. This characteristic suggests the potential for APG oral administration.

The study demonstrates that the dissolution behavior of APG is highly dependent on both the polymeric carrier and the pH of the environment. PLU127 consistently outperformed PLU68 across all tested conditions, showcasing its ability to improve APG solubilization through effective micellization and wettability enhancement. The biphasic release pattern of PLU127 highlights its potential for applications in controlled drug delivery systems. In contrast, PLU68 exhibited limited solubilization capabilities, with slow and linear release profiles across all conditions, making it unsuitable as a carrier for APG. Shaker et al. [47] indicate that the significant increase in the dissolution rate and the maximum amount of dissolved drug can be attributed to the direct solubilization effect of the amphiphilic carrier such as PLU. The authors suggest that PLU prevents the re-aggregation of drug crystals thanks to the impact of lowering surface and interfacial tension. The researchers base their conclusions on the results of tests conducted for solid dispersions of atorvastatin with PLU. Karolewicz et al. [48] indicate that the mechanism of increased dissolution rates of the drug from PLU’s dispersion could be related to polymer surface activity and wettability. This mechanism may have led to reduced drug agglomeration and thus increased the surface area and the solubilization effect of the carrier. Alshehri et al. [12] prepared the APG–PLU127 system using milling, melting, and microwave irradiation to compare these methods. Each method had three systems with different APG-to-carrier ratios: 1:1, 1:2, and 1:4. In all cases, a certain solubility level in water was reached after 15 min and was maintained up to 120 min. In the case of milling, the highest result was achieved for the system with a ratio of 1:4, which was 84.13% [12]. In the present work, with a ratio of 1:9, the percentage of release was 47.0% after 360 min. The difference between these results may be due to the presence of sodium lauryl sulfate (SLS), which was added to the study in the cited article in the amount of 1% per 900 mL of water. As a substance reducing surface tension, it could facilitate the flow of water through the capillaries inside the system particles [49]. A similar release percentage, i.e., about 85% in water, was also achieved by scientists from Saudi Arabia. They studied the APG–PLU127 system obtained by spray drying. However, similarly to the previously cited work, SLS was also used at a concentration of 1% to increase the wettability of the formulation [14]. The cited works did not provide information on the release profiles in media with different pH. The literature confirms that the addition of SLS significantly affects the solubility and release of the compound from the dispersion [50,51].

The SEM analysis was performed to investigate the surface morphology of apigenin, Pluronic F-127, Pluronic F-68, and their dispersion (Figure 11). The results revealed distinct characteristics for each sample. The SEM image of APG (Figure 11a) displayed crystalline particles with irregular shapes and sharp edges, indicative of its intrinsic crystalline structure. The particle size varied significantly. The SEM images of the dispersion showed a significant change in morphology compared to the individual components. The high dissolution rate of APG–PLU127 in relation to APG–PLU68 may be related to the size of the dispersion particles. The smaller size of APG–PLU127 particles led to an increase in the contact area of the solid dispersion with the solvent, resulting in a higher dissolution rate of the poorly soluble APG. In addition, smooth surfaces caused a slow APG release from this dispersion.

The thermal analysis of APG indicated a melting point at 366.3 °C (Figure 12, dashed line), and thermogravimetric (TG) results confirmed the stability of both APG and APG–PLU dispersions up to approximately 380 °C.

To further investigate the thermal properties of these dispersions, DSC measurements were conducted over a temperature range from 30 °C to 380 °C.

The results of the DSC analysis are presented in Figure 13. As previously mentioned, the DSC thermogram of APG exhibits an endothermic effect observed at about 366.3 °C, the temperature corresponding to the compound’s melting point (T_m_). The T_m_ values of PLU68 and PLU127 were observed at 57.5 °C and 59.7 °C, respectively, in agreement with literature data [52]. The DSC curves of physical mixtures and APG–PLU dispersions display only a single thermal effect, characteristic of pure PLU, with no peak from APG observed. Similarly, Agafonov et al. [53], in their study on methotrexate with Pluronic^®^ F127, also did not detect a characteristic peak for the API. They attributed this to the dissolution of the crystalline form of methotrexate in the molten carrier during preparation, resulting in the formation of a molecular solution. This explanation could also apply to the absence of the APG peak in our study, suggesting the formation of a molecularly dispersed solution. It is also noteworthy that the T_m_ of PLU shifted toward a lower temperature in the APG–PLU dispersions. The T_m_ values were recorded at 54.8 °C and 56.2 °C for APG–PLU68 BM and APG–PLU127 BM, respectively. No T_m_ shift was observed for PLU in the physical mixtures. This indicates the presence of chemical interactions between APG and PLU127/PLU68 in the obtained dispersions.

FT-IR analysis offers valuable insights into the structural properties of the analyzed material, allowing for the identification of specific bonds and chemical interactions formed between the compound and the excipient. The assignment of APG peak data to individual vibration modes in the molecule was developed based on the literature and is presented in Appendix A [54]. Figure 14 shows the FT-IR spectra in the range of 400–1800 cm^−1^ of APG, PLU68, a physical mixture (APG–PLU68 PM), and a solid dispersion (APG–PLU68 BM).

In the spectrum of pure PLU68, single peaks can be observed around 841 cm^−1^, 947 cm^−1^, 962 cm^−1^ (stretching vibrations of C-O bonds), 1061 cm^−1^, 1099 cm^−1^ (asymmetric stretching of ether groups), 1146 cm^−1^ (vibrations of CH_2_ groups), 1242 cm^−1^ (C–O–C modes), 1281 cm^−1^ (C–O–C modes), 1342 cm^−1^ (bending vibrations of the OH groups), 1360 cm^−1^ (methylene wagging modes), and 1468 cm^−1^ (methylene scissor modes) [39,48,55,56]. In the spectrum of the physical mixture of APG and PLU68, cumulative peaks originating from both substances are visible. After the milling process, the PLU68 peaks are still present in the spectrum, while most of the APG peaks are no longer visible, except for those at 428 cm^−1^ and 1032 cm^−1^, whose intensity decreased significantly. Some of the APG peaks also shifted: from 501 cm^−1^ to 505 cm^−1^, from 571 cm^−1^ to 581 cm^−1^, from 737 cm^−1^ to 743 cm^−1^, from 827 cm^−1^ to 829 cm^−1^, from 1557 cm^−1^ to 1558 cm^−1^, from 1605 cm^−1^ to 161 cm^−1^ and from 1651 cm^−1^ to 1657 cm^−1^. Their intensity also decreased, which indicates the dispersion of APG in the polymer. Moreover, changes in the nature of the spectrum indicate the formation of interactions between APG and PLU68. The disappearance of the peaks bending the phenolic groups may suggest the formation of a hydrogen bond with the strongly electronegative ether oxygen atom in PLU68 molecules. Figure 15 shows the FT-IR spectra in the range of 400–1800 cm^−1^ of APG, PLU127, a physical mixture (APG–PLU127 PM), and a solid dispersion (APG–PLU127 BM). The spectrum of PLU127 shows peaks around 841 cm^−1^, 962 cm^−1^ (stretching vibrations of C-O bonds), 1061 cm^−1^, 1099 cm^−1^ (asymmetric stretching of ether groups), 1146 cm^−1^, 1242 cm^−1^, 1281 cm^−1^ (vibrations of CH_2_ groups), 1342 cm^−1^ (bending vibrations of OH groups), 1360 cm^−1^ (methylene wagging modes), and 1468 cm^−1^ (methylene scissor modes) [39,48,55,56]. All of them appear in the spectrum of physical mixture together with peaks originating from APG. After the milling process, all peaks of PLU127 are still present in the spectrum, with the same intensity, while most of the APG peaks are no longer visible except those at 428 cm^−1^, 579 cm^−1^, 806 cm^−1^, 827 cm^−1^, 908 cm^−1^, 1032 cm^−1^, 1557 cm^−1^, and 1651 cm^−1^, the intensity of which decreased significantly. There are also APG peaks that have shifted: from 501 cm^−1^ to 505 cm^−1^, from 737 cm^−1^ to 741 cm^−1^, from 1177 cm^−1^ to 1182 cm^−1^, and from 1606 cm^−1^ to 1609 cm^−1^, whose intensity also decreased.

This indicates the molecular dispersion of the tested substance in the polymer matrix and the formation of molecular interactions. Similarly to PLU68, the formation of hydrogen bonds between the phenolic groups of APG and the oxygen atoms of PLU127 can be observed.

The analysis of XRPD was conducted to assess the stability of the APG–PLU sample under uncontrolled (ambient) conditions. This examination aimed to compare the structure of samples over time. The samples, initially analyzed immediately after preparation (time T0), were re-evaluated one year later (time T1) (Figure 16). The diffraction patterns at T1 revealed additional peaks attributed to the crystalline form of APG, which were absent in the T0 measurements. These observations suggest a partial recrystallization of APG over the storage period, indicating changes in the physical state of the material. Such results may point to limitations in the stability of the sample under ambient conditions, as the recrystallization of APG could impact its solubility.

To exclude the possibility of degradation products forming during storage, the APG–PLU68 and APG–PLU127 analyzed at T1 were subjected to high-performance liquid chromatography (HPLC). The obtained chromatograms were carefully compared with those of the sample examined immediately after preparation (T0) (Figure 17).

This analytical approach aimed to ensure the chemical stability of the solid dispersions over the one-year storage period under ambient conditions. The HPLC results demonstrated no significant differences in the chromatographic profiles between the T0 and T1 samples. Specifically, no additional peaks corresponding to degradation products were detected, confirming that the chemical integrity of the sample remained intact. This finding suggests that despite the observed recrystallization of APG noted in the XRPD analysis, the molecular structure of the sample’s components remained unchanged over time.

The confirmation of chemical stability at T1 is an essential outcome, as it highlights the robustness of the formulation in terms of preventing chemical degradation under ambient conditions. Such stability is crucial for maintaining the therapeutic efficacy and safety of the product during storage. These results collectively indicate that while the sample experienced physical changes, its chemical stability was successfully preserved, reflecting the formulation’s resilience against potential degradation processes.

The solubility of the APG–PLU127 sample was further evaluated by comparing the T0 and T1 samples after 4 h of incubation in water at 37 °C. This specific incubation period was chosen based on the release profile of APG from the APG–PLU127 solid dispersion, as it represents the time point at which the maximum percentage of the active substance was released, reaching a plateau. By analyzing the dissolution behavior under these controlled conditions, the aim was to determine whether the recrystallization observed in the XRPD analysis had an impact on the solubility of the sample.

HPLC analysis of the solubility revealed that the concentration of APG in solution after 4 h was 252 ± 1 μg∙mL^−1^ for T0 and 246 ± 1 μg∙mL^−1^ for T1 (statistically significant difference, *p* < 0.05). These results indicate that the physical changes observed over the one-year storage period did affect the dissolution performance of the solid dispersion. Although a difference in solubility between the samples was observed, it was minimal, amounting to only 6 μg∙mL^−1^.

Our previous studies [43,57,58] confirm that the improvement of water solubility translates into improved biological properties compared to the unmodified compound. For this reason, an antioxidant activity test was conducted using the 2,2′-azino-bis-(3-ethylbenzothiazoline-6-sulfonic) acid test (ABTS test) for APG–PLU127 T0 and T1 aqueous solutions (containing ~50 mg of dispersion in 5 mL of water). The results of the study confirmed that the aqueous solution of APG–PLU127 T0 showed 68.1% ± 1.94% ABTS scavenging ability, whereas APG–PLU127 T1 showed 66.2% ± 1.62% (statistically significant difference, *p* < 0.05). Nevertheless, activity remained high and correlated with the solubility of the sample.

## 3. Materials and Methods

### 3.1. Materials

The following reagents were used in the experiments: apigenin (Sigma Aldrich, St. Louis, MO, USA)), ethanol (POCH, Gliwice, Poland)), acetonitrile (HPLC grade, POCH, Gliwice, Poland), 96% formic acid (POCH, Gliwice, Poland), 0.1 N hydrochloric acid (Alfa Chem sp. z o.o., Poznan, Poland), potassium dihydrogen phosphate (POCH, Gliwice, Poland), sodium hydroxide (POCH, Gliwice, Poland), polyvinylpyrrolidone (PVP30, BASF SE, Ludwigshafen am Rhein, Germany), copolymer of vinylpyrrolidone and vinyl acetate (PVPVA64, BASF SE, Ludwigshafen am Rhein, Germany), sodium alginate (SA, Sigma Aldrich, St. Louis, MO, USA), Pluronic^®^ F-68 (PLU68, BASF SE, Ludwigshafen am Rhein, Germany), and Pluronic^®^ F-127 (PLU127, Sigma Aldrich, St. Louis, MO, USA).

### 3.2. Preparation of Solid Dispersion of Apigenin

Solid dispersions were obtained by ball milling. The APG–PLU physical mixtures (APG–PLU PM, 10% content of APG) were ground in a mortar. Next, APG–PLU PM was placed in a 50 mL stainless steel jar with three stainless steel balls (diameter = 12 mm). The milling process was carried out at room temperature (frequency: 30 Hz, time: 60 min). For further research, the powders were kept in a desiccator.

### 3.3. Identification of Neat Compounds and Solid Dispersions

#### 3.3.1. X-ray Powder Diffraction (XRPD)

Using a Bruker D2 Phaser diffractometer (Bruker, Germany), powder X-ray diffractometry was used to confirm the APG’s physical state for the following samples: (i) the pure compound, (ii) the physical mixture, and (iii) solid dispersion. CuKα radiation (1.54060 Å) at tube voltages of 30 kV and tube currents of 10 mA was used to record the diffraction patterns. The angular range was 5° to 40° 2θ with a step size of 0.02° 2θ and a counting rate of 2 s·step^−1^. Origin 2021b software (OriginLab Corporation, Northampton, MA, USA) was used to evaluate the acquired data.

#### 3.3.2. Scanning Electron Microscopy (SEM)

SEM images were captured using a scanning electron microscope Mira-3 Tescan (Tescan, Brno, Czech Republic) to assess the morphology of the samples. Before analysis, the samples were sputter-coated with carbon. The diameters of apigenin particles were measured using the ImageJ program (https://imagej.net/nih-image/, accessed on 19 December 2024).

#### 3.3.3. TG and DSC Study

Thermogravimetric (TG) analysis was conducted using a TG 209 F3 Tarsus^®^ micro-thermobalance (Netzsch, Selb, Germany). Approximately 10 mg of powdered samples were placed in an open Al_2_O_3_ crucible (85 µL) and heated at a rate of 10 °C·min^−1^ from 25 °C to 400 °C under a nitrogen atmosphere with a flow rate of 250 mL·min^−1^. The TG data were processed with Proteus 8.0 software (Netzsch, Selb, Germany), while the results were visualized using Origin 2021b software (Origin Lab Corporation, Northampton, MA, USA).

Thermal analysis was carried out using a DSC 214 Polyma differential scanning calorimeter (Netzsch, Selb, Germany), with a blank aluminum DSC pan serving as the reference. Approximately 10 mg of powdered samples were placed in pans with pierced covers. A single heating mode was applied to (i) determine the melting point of PLU in its pure form and in dispersions (temperature range: 25–380 °C, scanning rate: 10 °C·min^−1^) and (ii) determine the melting point of APG in its pure form (temperature range: 25–390 °C, scanning rate: 10 °C·min^−1^).

The analysis was performed in a nitrogen atmosphere at a flow rate of 250 mL·min^−1^. Data obtained from the DSC measurements were processed using Proteus 8.0 software (Netzsch, Selb, Germany), and the results were visualized with Origin 2021b software (OriginLab Corporation, Northampton, MA, USA).

#### 3.3.4. ATR-FTIR Spectroscopy

Using an IRTracer-100 spectrophotometer with a QATR that included a diamond ATR system (Shimadzu, Kyoto, Japan), ATR-FTIR spectra in the MIR range (400–1800 cm^−1^) were acquired with a spectral resolution of 4 cm^−1^. Signals for 100 scans were averaged for a single spectrum. All infrared spectra were gathered using LabSolution IR software (version 1.86 SP2, Shimadzu, Kyoto, Japan). The obtained data were visualized using Origin 2021b software (OriginLab Corporation, Northampton, MA, USA).

### 3.4. Studies of Results Introduction of Apigenin into Solid Dispersion

#### 3.4.1. HPLC Studies of Changes of Apigenin Concentrations

Changes in apigenin (APG) concentrations were analyzed using HPLC with a diode array detector (Shimadzu Corp., Kyoto, Japan). A ReproSil-Pur Basic-C18 column (100 × 4.6 mm; 5 µm, Dr. Maisch) was used as the stationary phase. Mobile phase A (45%): water + 0.1% formic acid; mobile phase B (55%): acetonitrile. The mobile phase flow rate was set at 0.5 mL∙min^−1^, with a detection wavelength of 336 nm. The injection volume was 10 µL, and the column temperature was maintained at 30 °C.

#### 3.4.2. Preparation of Media for Solubility Studies and Dissolution

High-purity laboratory-grade water was produced using a Direct-Q 3 UV purification system (model Exil SA 67120, Millipore, Molsheim, France). The hydrochloric acid solution at pH 1.2 was prepared according to the manufacturer’s (Alfachem, Poznan, Poland) instructions using an analytically weighed amount of the reagent. To prepare a buffer solution at pH 5.5, 750 mL of 0.1 M sodium acetate solution was placed in a 1000 mL volumetric flask. Then, 250 mL of 0.1 M acetic acid solution was added. The volume was adjusted to 1000 mL with purified water. A phosphate buffer at pH 6.8 was prepared by transferring 250 mL of a 0.2 N potassium dihydrogen phosphate solution into a 1000 mL volumetric flask. Subsequently, 112 mL of a 0.2 N sodium hydroxide solution was added, and the mixture was diluted to 1000 mL with purified water.

#### 3.4.3. Apparent Solubility Studies

In this experiment, excess APG and solid dispersions were placed in glass vials along with 4 mL of distilled water or buffer that was prepared in accordance with the procedure presented in Section 3.4.2. The samples were incubated in a MaxQ 4450 laboratory incubator (Thermo Scientific, Waltham, MA, USA) at 298.15 K for 15 and 60 min, with a constant rotation speed of 75 rpm. After incubation, the suspensions were filtered using a 0.22 µm filter and subjected to HPLC analysis. Each measurement was performed three times.

#### 3.4.4. Dissolution Rate Studies

The release profile study was performed by modifying the method proposed by Mesallati et al. [59]. Four 100 mL glass beakers were filled with 30 mL of medium (distilled water and pH 1.2, pH 5.5, and pH 6.8 buffers), previously heated to 37 °C in an ES-20 incubator (Biosan). The beakers with the solutions were then placed on an RCT 5 digital magnetic stirrer (IKA) with the following process parameters: temperature of 37 °C, and 200 rpm.

A total of 100 mg of APG and APG–PLU68 and APG–PLU127 solid dispersions were weighed. The weighed samples were poured into separate beakers. A total of 1 mL of the solution was taken from the beakers at specific time points (1, 5, 10, 15, 20, 30, 60, 90, 120, 180, 240, and 300 min) and filtered through a nylon filter with a pore diameter of 22 μm for chromatographic vials. Each time the test sample was taken, 1 mL of the appropriate medium was added.

#### 3.4.5. Physical Stability

Powdered samples of APG–PLU solid dispersions were placed into 2 mL Eppendorf tubes and stored under ambient conditions for one year. XRPD was used to analyze the solid-state structures of the samples after storage. Additionally, HPLC was employed to assess their chemical stability.

#### 3.4.6. Antioxidant Activity

The ABTS assay (2,2′-Azino-bis(3-ethylbenzthiazoline-6-sulfonic acid)) was used to measure the antioxidant activity of water solutions of APG–PLU127 BM (T0) and APG–PLU127 BM (T1). Preparation of APG–PLU127 (T0) and (T1) water solutions: approximately 50 mg of the sample was weighed into a glass vial, and 5 mL of water was added. The sample was incubated for 4 h (based on the APG release results) in a MaxQ 4450 laboratory incubator (Thermo Scientific, Waltham, MA, USA) at 298.15 K with a constant rotation speed of 75 rpm. After incubation, the suspensions were filtered using a 0.22 µm nylon filter and subjected to ABTS analysis. The experiment was conducted on a 96-well plate, with spectrophotometric analysis of the samples. The method adhered to a previously published protocol [60]. Specifically, 25.0 µL of each sample was combined with 175.0 µL of ABTS solution, while the blank contained 175.0 µL of ABTS solution and water. The samples were incubated in the dark for 30 min at room temperature with continuous shaking. Absorbance was measured at 734 nm using a Multiskan GO plate reader (Thermo Fisher Scientific, Waltham, MA, USA).

## 4. Conclusions

In this study, five solid dispersions of apigenin (APG) were successfully prepared using ball milling. The dispersions were subjected to an apparent solubility study, which identified the APG–PLU68 and APG–PLU127 dispersions as the most effective based on their significant solubility improvements after 60 min in various media. DSC and FT-IR analysis revealed molecular interactions between apigenin and the polymer matrix in these formulations, indicating successful incorporation and stabilization of the active compound within the carriers. Among these, the APG–PLU127 system demonstrated exceptional performance, achieving 100% solubility at pH 6.8. This suggests its promising potential for use in drug formulations targeting physiological environments with similar pH values, such as the small intestine. After one year of storage under ambient conditions, APG–PLU127 solid dispersion exhibited physical instability (recrystallization confirmed by XRPD) and chemical stability (confirmed by HPLC). Further study revealed a slight decrease in the solubility of APG and its antioxidant activity.

The results highlight the critical role of the polymer carrier’s structure in determining the solubility, release profile, and functional stability of the active compound. PLU127, with its optimized hydrophilic–lipophilic balance and stronger interactions with APG, outperformed PLU68 in solubilizing and stabilizing the bioactive compound. This underscores the importance of selecting polymers with suitable hydrophilic and hydrophobic properties to enhance the performance of solid dispersions.

In summary, the APG–PLU127 dispersion emerges as a promising candidate for pharmaceutical applications due to its excellent solubility at relevant pH levels and high antioxidant activity. Future research should focus on (i) improving the physical stability of the obtained dispersion, (ii) scaling up formulation, and (iii) exploring its behavior in in vivo models to confirm its suitability for drug delivery systems. Additionally, further studies could investigate the impact of environmental factors such as temperature and humidity on the stability and efficacy of such dispersions to ensure their robustness across various storage conditions.

## Figures and Tables

**Figure 1 ijms-26-00566-f001:**
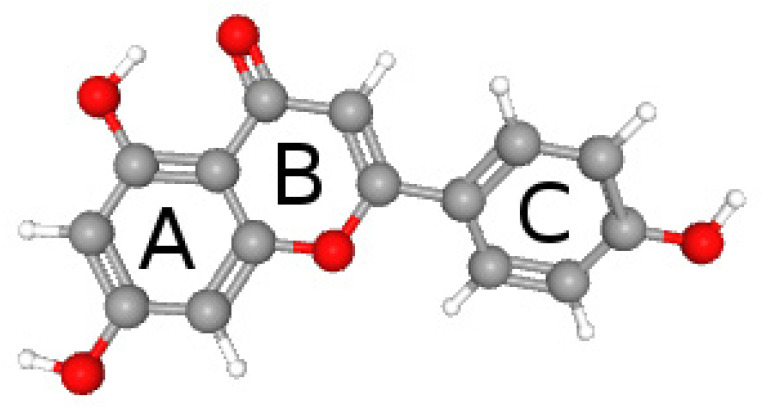
Structure of apigenin. A, B, and C—ring.

**Figure 2 ijms-26-00566-f002:**
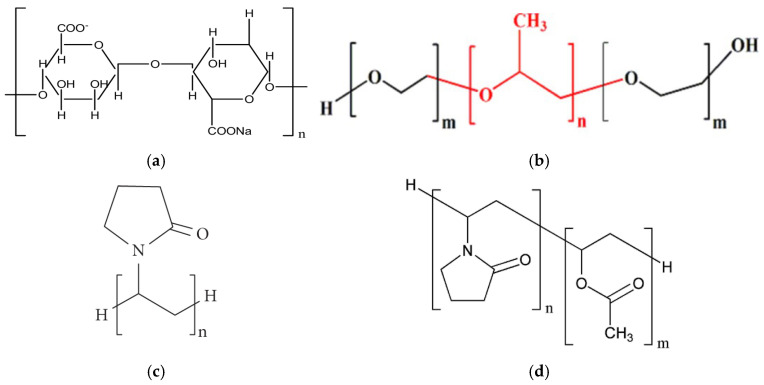
Structure of (**a**) sodium alginate (SA); (**b**) Pluronic^®^ (PLU): Pluronic^®^ F-68 (m = 77, n = 29) where red is propylene oxide part and black ethylene oxide part, Pluronic^®^ F-127 (m = 100, n = 65); (**c**) PVP K30 (PVP30); (**d**) PVP VA64 (PVPVA64).

**Figure 3 ijms-26-00566-f003:**
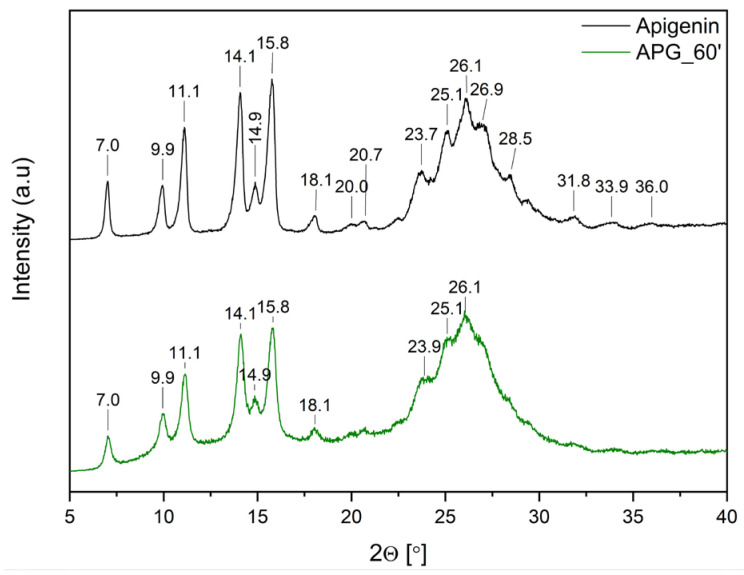
XRPD analysis: diffractogram of apigenin (APG, black), apigenin after 60 min of milling process (APG_60’, green).

**Figure 4 ijms-26-00566-f004:**
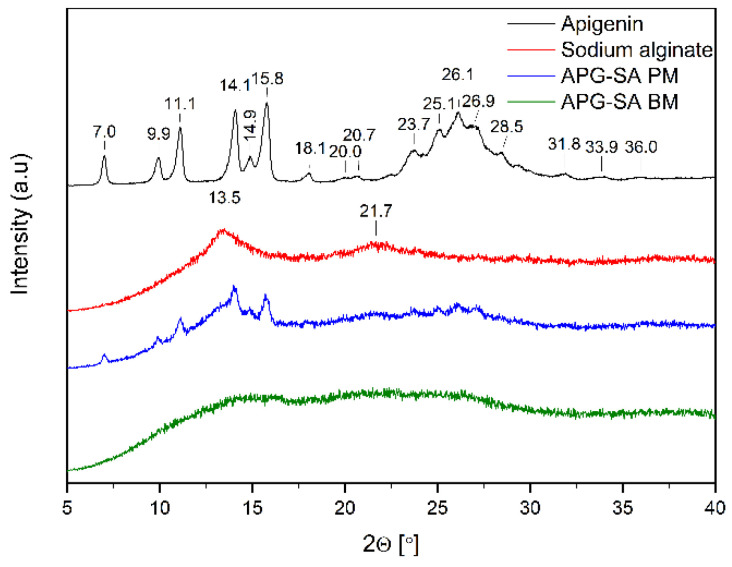
XRPD analysis: diffractogram of apigenin (APG, black), sodium alginate (SA, red), physical mixture of apigenin and sodium alginate (APG–SA PM, blue), apigenin and alginate SD (APG–SA BM, green).

**Figure 5 ijms-26-00566-f005:**
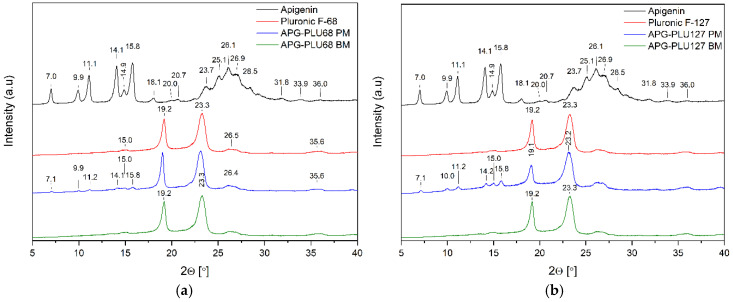
XRPD analysis: (**a**) diffractogram of apigenin (APG, black), Pluronic^®^ F-68 (PLU68, red), physical mixture of apigenin and Pluronic^®^ F-68 (APG–PLU68 PM, blue), apigenin and Pluronic^®^ F-68 solid dispersion (APG–PLU68 BM, green); (**b**) XRPD analysis: diffractogram of apigenin (APG, black), Pluronic^®^ F-127 (PLU127, red), physical mixture of apigenin and Pluronic^®^ F-127 (APG–PLU127 PM, blue), apigenin and Pluronic^®^ F-127 solid dispersion (APG–PLU127 BM, green).

**Figure 6 ijms-26-00566-f006:**
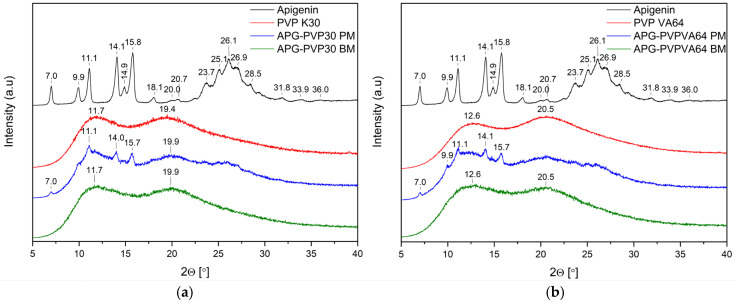
(**a**) XRPD analysis: diffractogram of apigenin (APG, black), PVP K30 (PVP K30, red), physical mixture of apigenin and PVP K30 (APG–PVP30 PM, blue), apigenin and PVP K30 solid dispersion (APG–PVP30 BM, green); (b) XRPD analysis: diffractogram of apigenin (APG, black), PVP VA64 (PVP VA64, red), physical mixture of apigenin and PVP VA64 (APG–PVPVA64 PM, blue), apigenin and PVP VA64 solid dispersion (APG–PVPVA64 BM, green).

**Figure 7 ijms-26-00566-f007:**
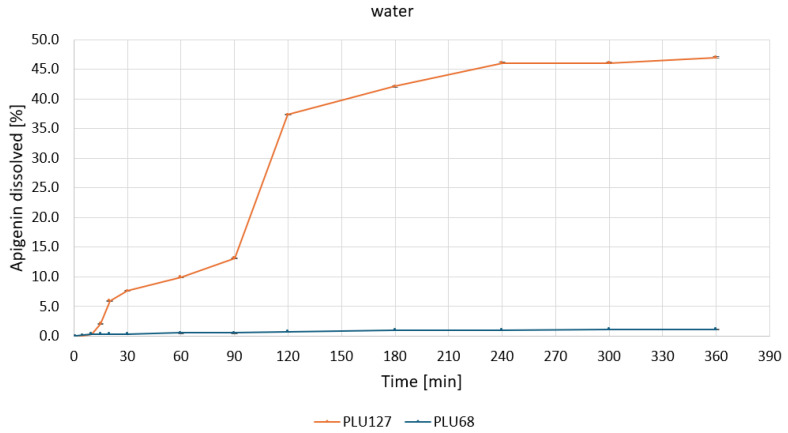
Apigenin release profiles in water from APG–PLU68 (blue line) and APG–PLU127 (orange line) solid dispersion.

**Figure 8 ijms-26-00566-f008:**
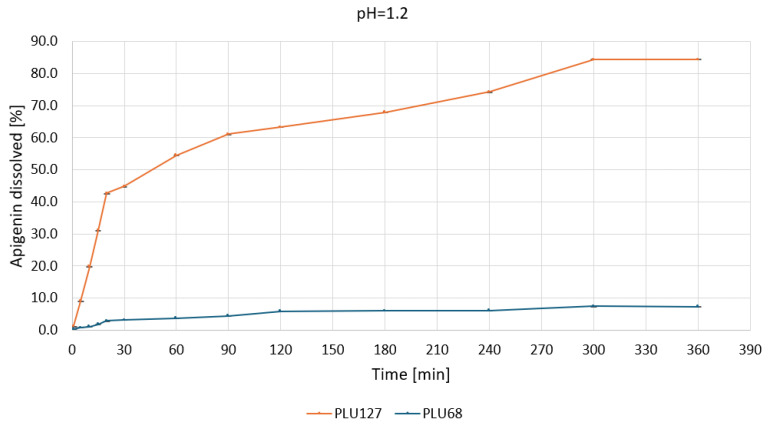
Apigenin release profiles in pH 1.2 from APG–PLU68 (blue line) and APG–PLU127 (orange line) solid dispersion.

**Figure 9 ijms-26-00566-f009:**
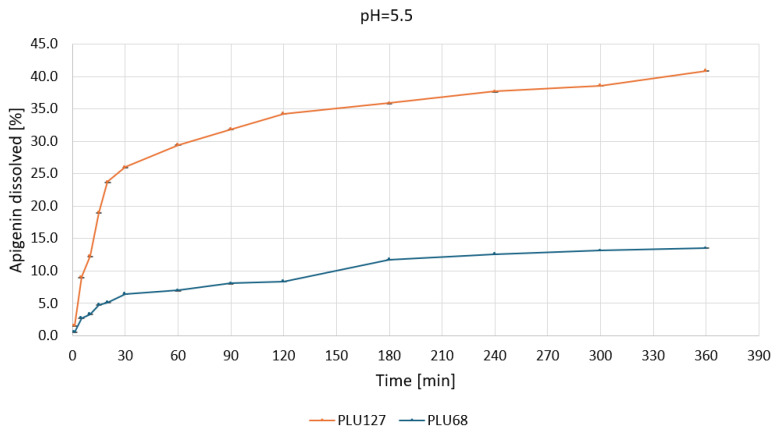
Apigenin release profiles in pH 5.5 from APG–PLU68 (blue line) and APG–PLU127 (orange line) solid dispersion.

**Figure 10 ijms-26-00566-f010:**
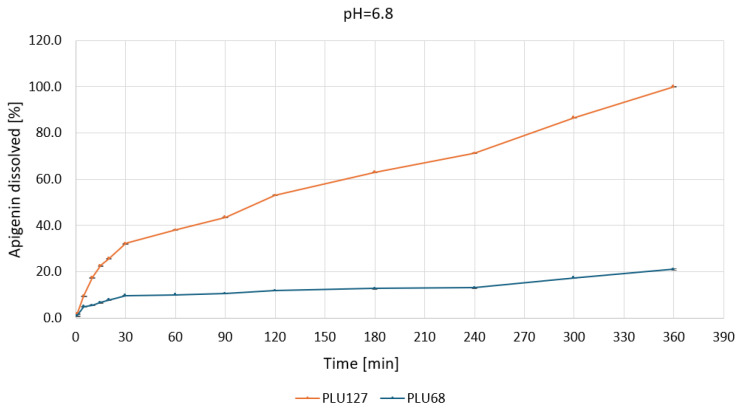
Apigenin release profiles in pH 6.8 from APG–PLU68 (blue line) and APG–PLU127 (orange line) solid dispersion.

**Figure 11 ijms-26-00566-f011:**
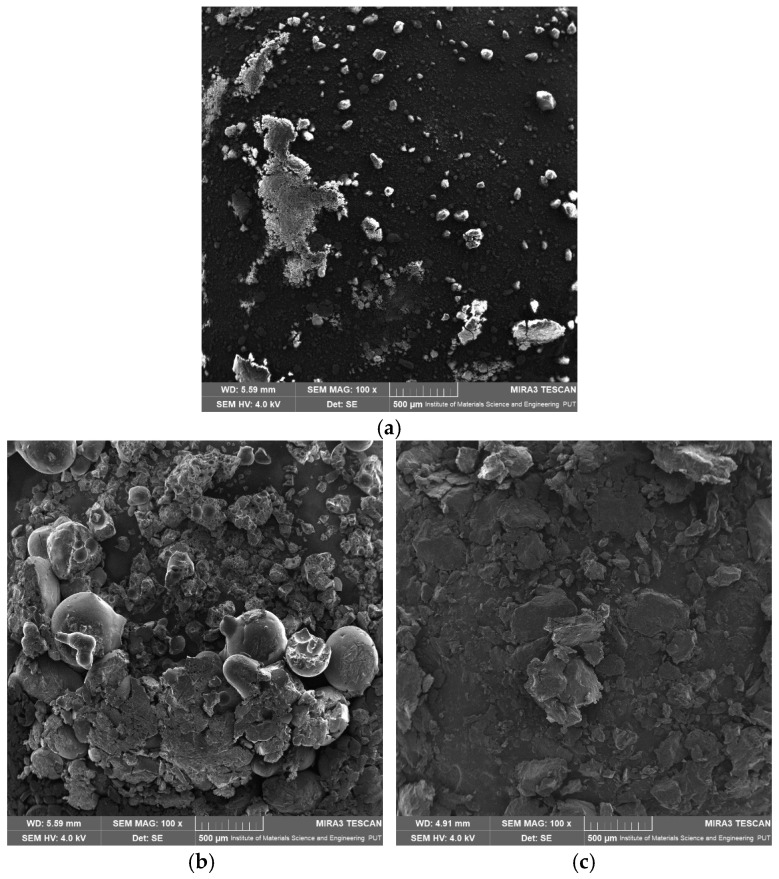
SEM images: (**a**) apigenin, (**b**) Pluronic F-127, (**c**) Pluronic F-68, (**d**) solid dispersion of apigenin–Pluronic F-127, (**e**) solid dispersion of apigenin–Pluronic F-68.

**Figure 12 ijms-26-00566-f012:**
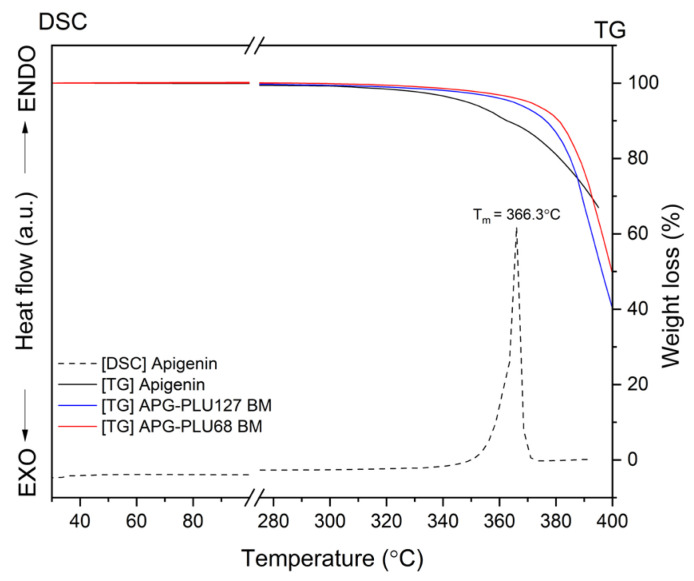
TG analysis: apigenin, APG–PLU127 solid dispersion, APG–PLU68 solid dispersion, and DSC analysis: apigenin.

**Figure 13 ijms-26-00566-f013:**
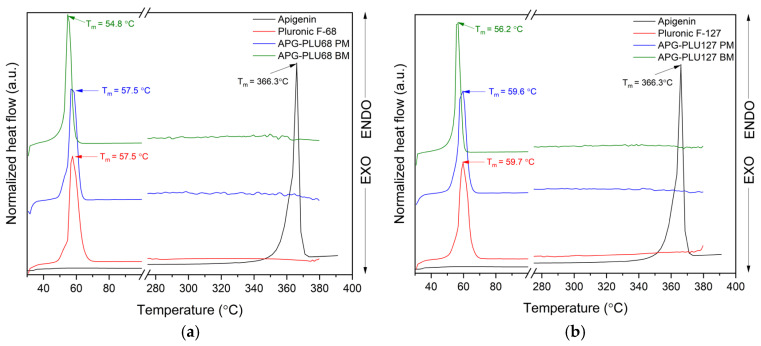
DSC analysis: (a) apigenin (APG, black), Pluronic^®^ F-68 (PLU68, red), physical mixture of apigenin and Pluronic^®^ F-68 (APG–PLU68 PM, blue), apigenin and Pluronic^®^ F-68 solid dispersion (APG–PLU68 BM, green); (**b**) DSC analysis: apigenin (APG, black), Pluronic^®^ F-127 (PLU127, red), physical mixture of apigenin and Pluronic^®^ F-127 (APG–PLU127 PM, blue), apigenin and Pluronic^®^ F-127 solid dispersion (APG–PLU127 BM, green).

**Figure 14 ijms-26-00566-f014:**
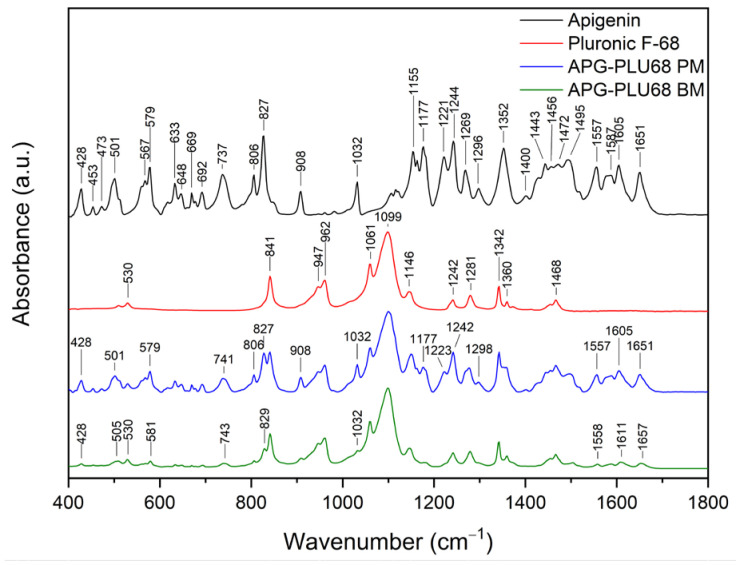
FT-IR analysis: apigenin (APG, black), Pluronic^®^ F-68 (PLU68, red), physical mixture of apigenin and Pluronic^®^ F-68 (APG–PLU68 PM, blue), apigenin and Pluronic^®^ F-68 solid dispersion (APG–PLU68 BM, green).

**Figure 15 ijms-26-00566-f015:**
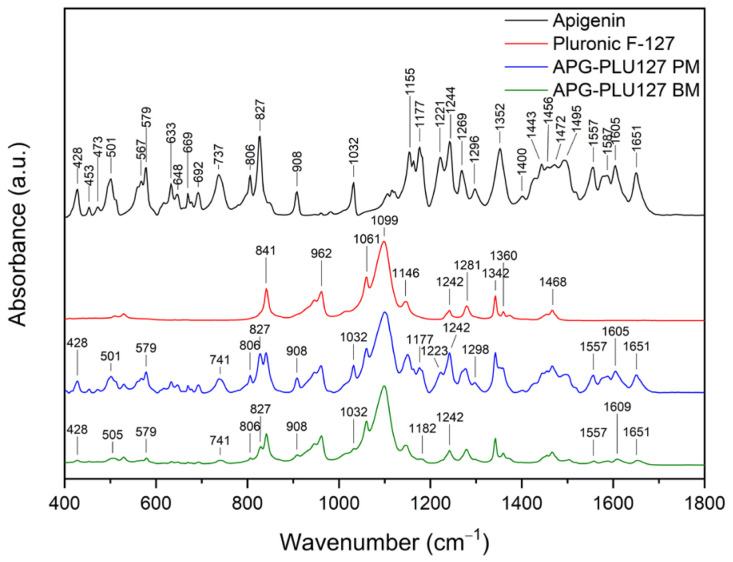
FT-IR analysis: apigenin (APG, black), Pluronic^®^ F-127 (PLU127, red), physical mixture of apigenin and Pluronic^®^ F-127 (APG–PLU127 PM, blue), apigenin and Pluronic^®^ F-127 solid dispersion (APG–PLU127 BM, green).

**Figure 16 ijms-26-00566-f016:**
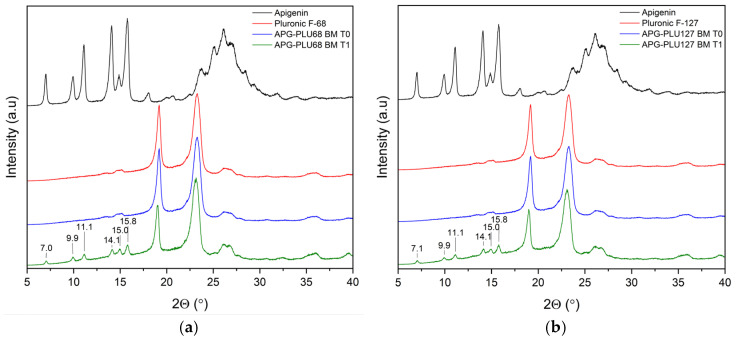
XRPD analysis: (**a**) diffractogram of apigenin (APG, black), Pluronic^®^ F-68 (PLU68, red), apigenin and Pluronic^®^ F-68 solid dispersion—fresh sample (APG–PLU68 BM T0, blue), apigenin and Pluronic^®^ F-68 solid dispersion—sample stored 1 year in ambient condition (APG–PLU68 BM T1, green); (**b**) XRPD analysis: diffractogram of apigenin (APG, black), Pluronic^®^ F-127 (PLU127, red), apigenin and Pluronic^®^ F-127 solid dispersion—fresh sample (APG–PLU127 BM T0, blue), apigenin and Pluronic^®^ F-127 solid dispersion—sample stored 1 year in ambient condition (APG–PLU127 BM T1, green).

**Figure 17 ijms-26-00566-f017:**
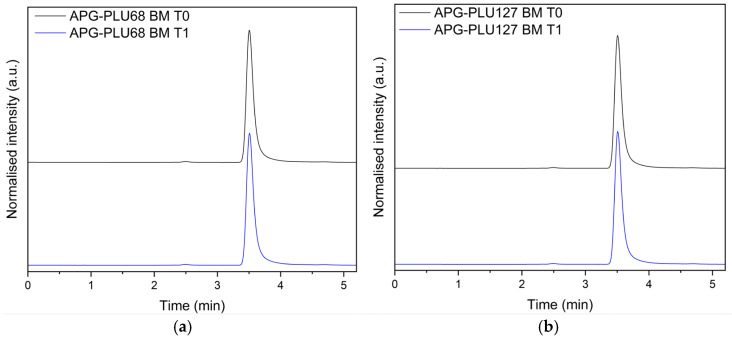
HPLC analysis: (**a**) apigenin and Pluronic^®^ F-68 solid dispersion—fresh sample (APG–PLU68 BM T0, blue), apigenin and Pluronic^®^ F-68 solid dispersion—sample stored 1 year in ambient condition (APG–PLU68 BM T1, black); (**b**) apigenin and Pluronic^®^ F-127 solid dispersion—fresh sample (APG–PLU127 BM T0, blue), apigenin and Pluronic^®^ F-68 solid dispersion—sample stored 1 year in ambient condition (APG–PLU127 BM T1, black).

**Table 1 ijms-26-00566-t001:** APG delivery systems are described in the literature.

Delivery System	Carrier	Reference
Solid dispersions	Pluronic^®^ F127, carbon nanopowder, silica nanoparticles, Soluplus^®^, Kollidon VA 64	[12,13,14,15,16]
Hydrogel	Chitosan, carbapol	[17,18,19]
Inclusion complexes with cyclodextrins (CD)	α-CD, β-CD, hydroxypropyl-β-CD, methyl-β-CD	[20,21]
Lipid carriers	Micelles composed of Pluronic^®^ P123 and Solutol HS 15	[22,23]
Liposomes	Distearoylphosphatidylcholine	[24,25]
Nanoparticles	Casein, PVP K30	[26,27]

**Table 2 ijms-26-00566-t002:** Apparent solubility study: APG concentration in water and buffers at pH 1.2, 5.5, and 6.8 after 15 and 60 min.

Carrier	15 min APG Concentration [μg∙mL^−1^]				60 min APG Concentration [μg∙mL^−1^]			
	water	pH 1.2	pH 5.5	pH 6.8	water	pH 1.2	pH 5.5	pH 6.8
none	x	x	x	x	x	x	x	x
PLU68	1.99 ± 0.02	0.59 ± 0.04	0.39 ± 0.01	0.36 ± 0.00	0.54 ± 0.02	0.87 ± 0.00	21.37 ± 0.60	7.96 ± 0.08
PLU127	20.35 ± 0.01	24.47 ± 0.69	16.62 ± 0.01	40.98 ± 0.07	70.48 ± 0.31	147.11 ± 0.12	327.94 ± 0.53	296.40 ± 0.44
PVP30	x	0.09 ± 0.00	0.28 ± 0.01	x	x	0.03 ± 0.01	0.06 ± 0.01	x
PVPVA64	x	x	x	x	x	x	x	x
SA	x	x	x	x	x	x	x	x

x—APG concentration below the detection threshold.

## Data Availability

The data are contained within the article and Appendix A.

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
