# Peer review of "Enhancing the Solubility and Dissolution of Apigenin: Solid Dispersions Approach"

_ijms, 2025, doi:10.3390/ijms26020566_

Round 1

Reviewer 1 Report (Previous Reviewer 3)

Comments and Suggestions for Authors

The revised manuscript by Natalia Rosiak et al., titled “Enhancing the Solubility and Dissolution of Apigenin: Solid Dispersions Approach,” represents a significant improvement and demonstrates the authors' commendable dedication to addressing the concerns previously raised. This updated version reflects a marked enhancement in scholarly rigor, offering a more thorough and sophisticated analysis of the solubility and dissolution of apigenin through the solid dispersion approach. The expanded discussion, coupled with improved clarity and precision, significantly elevates the manuscript's academic quality, providing a comprehensive and insightful perspective on this critical topic. I am pleased to recommend acceptance of this revised submission, as it constitutes a valuable and impactful contribution to the field, advancing understanding and knowledge in this domain.

Author Response

Dear Reviewer,

We greatly appreciate your recognition of the improvements we have made in response to the concerns you valuable comments. We are grateful for your recommendation for acceptance. Once again, thank you for your time and expertise in reviewing our manuscript.

Reviewer 2 Report (Previous Reviewer 2)

Comments and Suggestions for Authors

The study presented by Rosiak et al. is an important contribution to the pharmaceutical sciences in that it explores the solubilization of Apigenin via the preparation of solid dispersions with different excipients using ball milling. The authors have addressed and modified a few comments from the previous review and improved upon their previous manuscript. The manuscript demonstrates a well-organized experimental design and presents data relevant for enhancing drug solubility, which is a highly critical challenge in drug formulation. Below are some specific comments and recommendations for improving the manuscript before publication:

The mobile phase consisted of 0.1% formic acid (45%) and acetonitrile (55%). Hopefully, the HPLC solvent system is biphasic, meaning it should contain other solvents besides acetonitrile. I previously mentioned the same concern, which I believe the authors failed to fully comprehend. I would like to provide clarification for the authors to enhance their comprehension. If one is using 0.1 % formic acid, then it should be dissolved either in methanol or water or any other solvent or buffer. I understand that some solvent is taken volume-wise at 45% (consisting of 0.1% formic acid) and acetonitrile is 55%. Authors should focus on the clarity of this line. I don’t recommend using the words “Phase A” and “Phase B”; instead, they can be written as solvent A and solvent B.

It does not make sense if one mentions a 1 N HCl buffer (pH 1.2), pH 6.8 buffer, and pH 5.5 buffer. Please specify which buffer is used here. —the lines were not corrected since the first revision. There are many forms of pH-adjusted buffer available. Mention a specific buffer in the method/materials section for the readers to fully understand.

I previously suggested using SEM (scanning electron microscope) to observe the surface topography of the samples. I have already discussed the importance of SEM analysis in my previous review report, but I would like to reiterate its importance here. SEM analysis should probe the morphology and size of the prepared samples. The surface area and shape of solid dispersions affect the rate at which formulations dissolve. SEM will provide valuable insights into the relationship between these physical properties and the effectiveness of the formulations. Rough surfaces suggest high dissolution rates, while smooth surfaces could indicate a slow drug release. That's why it's important to include this data in this manuscript. I would recommend incorporating the morphology, shape, porosity, and size of the solid dispersion through SEM analysis (At least for APG-PLU127 ).

Author Response

The study presented by Rosiak et al. is an important contribution to the pharmaceutical sciences in that it explores the solubilization of Apigenin via the preparation of solid dispersions with different excipients using ball milling. The authors have addressed and modified a few comments from the previous review and improved upon their previous manuscript. The manuscript demonstrates a well-organized experimental design and presents data relevant for enhancing drug solubility, which is a highly critical challenge in drug formulation. Below are some specific comments and recommendations for improving the manuscript before publication:
The mobile phase consisted of 0.1% formic acid (45%) and acetonitrile (55%). Hopefully, the HPLC solvent system is biphasic, meaning it should contain other solvents besides acetonitrile. I previously mentioned the same concern, which I believe the authors failed to fully comprehend. I would like to provide clarification for the authors to enhance their comprehension. If one is using 0.1 % formic acid, then it should be dissolved either in methanol or water or any other solvent or buffer. I understand that some solvent is taken volume-wise at 45% (consisting of 0.1% formic acid) and acetonitrile is 55%. Authors should focus on the clarity of this line. I don’t recommend using the words “Phase A” and “Phase B”; instead, they can be written as solvent A and solvent B.

The authors improved the description to make it clear and understandable.

It does not make sense if one mentions a 1 N HCl buffer (pH 1.2), pH 6.8 buffer, and pH 5.5 buffer. Please specify which buffer is used here. —the lines were not corrected since the first revision. There are many forms of pH-adjusted buffer available. Mention a specific buffer in the method/materials section for the readers to fully understand.

We sincerely thank you for your valuable comment. During the revision of the manuscript, the authors introduced a new point "3.4.2 Preparation of Media for Solubility Studies and Dissolution", therefore in the section "Apparent Solubility Studies" no changes were made. At the suggestion of the reviewer, the authors introduced additional changes aimed at improving the readability of the manuscript.

I previously suggested using SEM (scanning electron microscope) to observe the surface topography of the samples. I have already discussed the importance of SEM analysis in my previous review report, but I would like to reiterate its importance here. SEM analysis should probe the morphology and size of the prepared samples. The surface area and shape of solid dispersions affect the rate at which formulations dissolve. SEM will provide valuable insights into the relationship between these physical properties and the effectiveness of the formulations. Rough surfaces suggest high dissolution rates, while smooth surfaces could indicate a slow drug release. That's why it's important to include this data in this manuscript. I would recommend incorporating the morphology, shape, porosity, and size of the solid dispersion through SEM analysis (At least for APG-PLU127).

We are grateful for your valuable comments and suggestions. The authors performed the SEM analysis. The results and description were included in the main manuscript. SEM images are in Figure 11.

Round 2

Reviewer 2 Report (Previous Reviewer 2)

Comments and Suggestions for Authors

Rosiak et al. responded to all previously recommended comments. I am satisfied the progress they made. I would recommend address the following before the publication.

Double-check figure 11. There are 6 images, whereas the naming was done for 5 images.

In the “HPLC Studies of Changes of Apigenin Concentrations” section instead of writing the lines 762-769, Write the following line. 

Mobile phase A (45%): water + 0.1% formic acid, mobile phase B (55%): acetonitrile

Author Response

Comments 1 Response: We would like to thank the reviewer for appreciating our work. We would also like to thank you for your time and valuable comments.

Comments 2 Response: Thank you reviewer for pointing out figure 11. In the track changes mode, you can see 6 images (of which 1 is as "deleted", it was a duplicate figure "e"). The authors accepted the change in the manuscript so that the figure would not be misleading. The authors confirm that Figures are 5: (a) apigenin, (b) Pluronic F-127, (c) Pluronic F-68, (d) solid dispersion of apigenin-Pluronic F-127, (e) solid dispersion of apigenin-Pluronic F-68.

Comments 3 Response: The authors improved the text according to the reviewer's suggestion.

This manuscript is a resubmission of an earlier submission. The following is a list of the peer review reports and author responses from that submission.

Round 1

Reviewer 1 Report

Comments and Suggestions for Authors

The article addresses a relevant issue in pharmaceutical sciences, focusing on improving the bioavailability of apigenin, a bioactive flavonoid with multiple therapeutic benefits.

1. Present the chemical structure of apigenin in the Introduction section

2. The abstract does not fully summarize the key results of the study. It would benefit from including specific data points related to solubility improvements and release profiles

3. In the section presenting the results, especially the solubility study (Table 1), the discussion lacks clarity regarding the reasons behind the different behaviors of the solid dispersions. It would be helpful to explain more deeply why PLU127 showed significantly better solubility than PLU68.

4. The comparison with previous studies (lines 213–225) could benefit from clearer reasoning to explain the discrepancies in results between this study and the Alshehri study. Was sodium lauryl sulfate's presence the sole reason for the difference, or were there other potential factors?

5. Some figures (e.g., Figure 1) could be more user-friendly with larger font sizes on the axes and clearer labeling.

6. The description of the preparation of solid dispersions using the ball milling technique lacks detail about the frequency and duration of the process in comparison with other techniques like solvent evaporation or spray drying

Comments on the Quality of English Language

The quality of the English language in the manuscript is generally good

Reviewer 2 Report

Comments and Suggestions for Authors

In the current study, Rosiak et al. sought to present a comprehensive formulation technique entitled “Enhancing the Solubility and Dissolution of Apigenin: Solid Dispersions Approach.” In this research article, authors investigated the solid dispersions of APG prepared using ball milling with sodium alginate (SA), Pluronic F-68 (PLU68), Pluronic F-127 15 (PLU127), PVP K30, and PVP VA64 as excipients. These dispersions were screened for apparent solubility in water and at different pH ranges. Based on improved solubility after 60 minutes, APG-PLU127 was found suitable for further study.

In general, the work exhibits a commendable level of quality and coherence. However, before publication, it is important to address a few provided suggestions.

  • The introduction section should include the chemical structure of apigenin.
  • The first paragraph requires sufficient references. There is only one reference mentioned.
  • Cite previous work from other groups on the process of enhanced solubility and bioavailability of apigenin. What is the advantage of pursuing ball milling for improving the solubility and dissolution of apigenin compared to the other methods reported so far? Include in the introduction section.
  • Mention the apigenin's solubility in water.
  • Apigenin is considered an APG; please include this information in the introduction section.
  • Line 328: Mention the name of the filter paper with proper information.
  • I have not seen any HPLC Studies of Changes of Apigenin Concentrations. Please include this information in either the main manuscript or the Supplementary Information.
  • The mobile phase consisted of 0.1% formic acid (45%) and acetonitrile (55%). Hopefully, the HPLC solvent system is biphasic, meaning it should contain other solvents besides acetonitrile. Correct the corresponding line.
  • The conclusion section should be rewritten, as Line 339-346 only demonstrated what could be done with this formulation. I would recommend including information about the improvements made with the new formulation by the authors.
  • It does not make sense if one mentions 1 N HCl buffer (pH 1.2), pH 6.8 buffer, and pH 5.5 buffer. Please specify which buffer is used here.
  • The thermal properties of the solid dispersions would be ideal to include in this manuscript. The interactions between the drug and the polymer can affect thermal behavior. Thermal analysis can provide an indication of miscibility or potential phase separation, which is crucial for maintaining stability over time.
  • SEM analysis should probe the morphology and size of the prepared samples. The surface area and morphology of solid dispersions affect the dissolution rates of the formulations; SEM will provide valuable data for the correlation of these physical properties with the formulation's performance. High dissolution rates can be implied from a rough surface, whereas smooth surfaces may reflect a slow release of drugs. That's why it's important to include this data in this manuscript.
  • It is vital to understand if the APG-PLU127 solid dispersion shows good efficacy in vitro, and a comparative study with the raw apigenin with the APG-PLU127 is warranted for its practical use.

Comments on the Quality of English Language

English language is fine. Minor improvement is necessary.

Reviewer 3 Report

Comments and Suggestions for Authors

The manuscript submitted by Natalia Rosiak et al, titled “Enhancing the Solubility and Dissolution of Apigenin: Solid Dispersions Approach. While the authors have demonstrated notable scientific rigor and innovation, but addressing the following raised question would not only strengthen the overall quality of the work but also significantly enhance its scholarly impact, positioning it for well-deserved recognition and imminent acceptance.

Detailed Comments:

1.     While the study indicates the potential improvement in solubility, it lacks crucial insights into the long-term stability of these formulations and their scalability for commercial production. To assess their practical feasibility, additional data on the stability of the dispersions under various storage conditions (e.g., temperature, humidity) should also be considered. A key question remains whether the APG-PLU systems can maintain their solubility and dissolution rates over time, particularly as Pluronic F-127 is known to precipitate if refrigerated or frozen. Addressing this point would help evaluate the potential for broader applications.

2.     The APG-PLU68 and APG-PLU127 systems were selected for further analysis, while the other formulations (e.g., APG-SA, APG-PVP K30, APG-PVP VA64) were not equally promising. Were the other systems significantly inferior in terms of solubility or stability? The authors are suggested to clarify the reasons for selecting only two formulations would help substantiate the conclusions

3.     The use of Pluronic F-68 and Pluronic F-127 raises potential concerns about toxicity and side effects, particularly with prolonged or high-dose use. This is especially pertinent for delivery routes like buccal and sublingual, where long-term exposure could pose risks. A thorough safety evaluation, with data on the long-term effects of these excipients, would be necessary before advancing these formulations to therapeutic use, particularly for sensitive routes like buccal, sublingual, or rectal.

4.     While the APG-PLU68 and APG-PLU127 systems were evaluated for their release profiles, the study lacks specific data or comparative analysis of these profiles. A more detailed discussion of the release kinetics and efficiency is needed to fully support the claim that these formulations improve drug delivery.

5.     Although the dissolution rate improvements are promising, there is a need for a discussion or preliminary data on the in vivo bioavailability of these formulations. Enhanced solubility does not necessarily equate to better bioavailability, and pharmacokinetic evaluation would be critical to fully assess the therapeutic potential of the APG-PLU systems. The authors should consider addressing how these solubility enhancements translate into in vivo efficacy.

6.     The authors are encouraged to incorporate more recent references from relevant studies to provide a comprehensive and up-to-date context for their work. This will enhance the credibility of the research by aligning it with the latest advancements in the field.